# When AI Coding Assistants Leak Training Data: Study Memorization in Code LLMs

## Abstract

Large Language Models (LLMs) for code generation risk memorizing and reproducing sensitive training data, including licensed code and proprietary information. We investigate memorization behavior in state-of-the-art code LLMs using a two-stage attack pipeline combining membership inference and data extraction. We evaluate four models (StarCoder2-3B, StarCoder2-7B, Llama3-8B, and DeepSeek-R1-distilled-Llama-8B) on a custom dataset of 30,000+ Python files. Our results reveal memorization rates of 42-64%, with code-specialized models exhibiting higher rates than general-purpose models. Categorical analysis shows that repetitive content (license headers, documentation) is memorized at rates up to 70%, while complex code exhibits lower susceptibility. Notably, realistic code completion scenarios trigger unintentional memorization in 13-14% of cases, posing practical risks for AI coding assistants. We demonstrate that knowledge distillation reduces extraction rates by approximately 19%, offering a cost-effective mitigation approach. Our findings confirm that memorization persists in modern LLMs and is influenced more by training domain and content characteristics than by parameter count alone.

## CCS Concepts

• **Security and privacy** → *Privacy protections*; • **Computing methodologies** → Machine learning.

## Keywords

Large Language Models (LLMs), Code Memorization, Membership Inference Attack, Software Security and Privacy

**ACM Reference Format:**
Anonymous Author(s). 2018. When AI Coding Assistants Leak Training Data: Study Memorization in Code LLMs. In *Proceedings of Make sure to enter the correct conference title from your rights confirmation email (Conference acronym 'XX)*. ACM, New York, NY, USA, 5 pages. https://doi.org/XXXXXXX.XXXXXXX

## 1 Introduction

Large language models (LLMs) demonstrate remarkable proficiency in code-related tasks, including generation, summarization, and completion [1]. These models are typically trained on vast datasets from publicly available sources, including library databases, discussion forums, and GitHub repositories [2]. However, this extensive

training can lead to unintended memorization, where models reproduce verbatim segments from their training data [3]. This phenomenon poses serious risks, including privacy breaches through exposure of personal identifiable information (PII), intellectual property violations when licensed code is reproduced without attribution [4], and performance degradation due to overfitting [5].

The severity of memorization in code LLMs is particularly concerning. Recent studies have shown that models can extract sensitive information such as API keys, authentication tokens, and proprietary algorithms [4]. More alarmingly, even with benign prompts in realistic code completion scenarios, models may inadvertently reproduce memorized training sequences [6]. These risks are amplified as LLMs become increasingly integrated into developer workflows through tools like GitHub Copilot and other AI-assisted coding platforms.

Previous research has established pipelines combining membership inference attacks (MIA) and data extraction attacks to reveal and quantify memorization [7]. However, these studies have primarily focused on older models and have not systematically examined how memorization varies across different code categories or whether recent models exhibit similar vulnerabilities. Furthermore, while various mitigation strategies have been proposed, including dataset deduplication [8], differential privacy, and goldfish loss [9], each involves significant trade-offs in cost or performance. Knowledge distillation has emerged as a promising alternative that may reduce memorization while preserving model capabilities [10], but its effectiveness for code LLMs remains underexplored.

In this work, we investigate memorization behavior in state-of-the-art code LLMs and evaluate knowledge distillation as a mitigation approach. We address three research questions: (1) Does the existing memorization-revealing pipeline remain effective on state-of-the-art code LLMs? (2) How do content categories and prompt characteristics affect memorization in code LLMs? (3) Can knowledge distillation reduce memorization in code LLMs? We evaluate StarCoder2-3B, StarCoder2-7B, and Llama3-8B using a dataset of over 30,000 Python code sequences, categorizing them into five types (license, documentation, dictionaries, code logic, and testing). Our findings show memorization rates ranging from 42% to 64%, with license and documentation categories most susceptible. Realistic code completion prompts can trigger memorization at rates of 26-43%. Notably, a distilled version of Llama3-8B reduces extraction rates by approximately 20%, demonstrating knowledge distillation as a viable mitigation strategy.

The main contributions of this paper are:
- A systematic evaluation of memorization in state-of-the-art code LLMs using an established attack pipeline.
- An analysis of memorization patterns across five code categories and realistic usage scenarios.
- Empirical evidence that knowledge distillation can reduce memorization rates while maintaining model performance.

- A curated dataset and evaluation methodology for future memorization research in code LLMs.

**Paper organization.** The remainder of this paper is organized as follows: Section 2 reviews background on memorization and related work. Section 3 describes our research methodology and experimental setup. Section 4 presents our findings. Section 5 discusses threats to validity, and Section 6 concludes with future work directions.

## 2 Background and Related Works

**Memorization and Risks** Generative models often memorize verbatim segments of training data instead of generalizing patterns [11]. This behavior persists across architectures and is particularly severe in code models; for instance, thousands of memorized snippets have been extracted from state-of-the-art outputs [4]. Such memorization poses critical risks: *privacy leakage* of PII and API keys [3, 12], *intellectual property violations* of licensed code [4, 13], and *performance degradation* due to overfitting [5, 14]. Attackers can exploit this via Membership Inference Attacks (MIA) [15] and data extraction pipelines [7, 16]. Notably, code exhibits category-specific memorization patterns due to its structural predictability [17], and even benign prompts can trigger unintentional data leakage in realistic scenarios [6].

**Mitigation and Distillation** Existing defenses against memorization involve significant trade-offs. *Deduplication* reduces near-duplicates but cannot eliminate unique memorized sequences [8, 11]; *Differential privacy* often degrades model utility [18]; and *Goldfish loss* incurs training overhead [9]. Knowledge distillation, which transfers behavior from a teacher to a smaller student model [19], has emerged as a promising mitigation strategy [10, 20]. By forcing the student to learn generalized logic rather than memorizing specific examples, distillation potentially balances privacy and performance. However, its effectiveness specifically for code LLMs remains underexplored, which motivates our systematic evaluation.

## 3 Approach

To systematically investigate memorization, we adopt a two-stage adversarial attack pipeline—Membership Inference Attack (MIA) followed by Data Extraction [7], as illustrated in Figure 1.

**Dataset and Models.** We constructed a diverse evaluation corpus by sampling over 30,000 Python files from BigQuery's GitHub dataset [21]. From these, we extracted random 150-token sequences, each split into equal 50-token parts: *pre-prefix*, *prefix*, and *suffix* (ground truth). We evaluate three base models to assess the impact of architecture and training data: StarCoder2-3B, StarCoder2-7B (code-specialized) [22], and Llama3-8B (general-purpose) [23]. Furthermore, to evaluate mitigation strategies, we include a distilled variant: DeepSeek-R1-distilled-Llama-8B [24].

**Two-stage adversary attack** In *Stage 1 (Membership Inference Attack)*, we prompt each model with a 100-token context (pre-prefix + prefix). Sequences whose generated output exhibits high similarity to the original suffix are flagged as likely candidates for training data. In *Stage 2 (Data Extraction Attack)*, we perform a guided extraction on these candidates using only the 50-token *prefix* to test the model's ability to reproduce training data with minimal context. To quantify successful memorization across all experiments, we evaluate the extracted completions against the original ground-truth suffixes using Exact Match (EM) and BLEU-4 scores.

## 4 Results

### RQ1: Does the existing memorization-revealing pipeline remain effective on state-of-the-art code LLMs?

To establish baseline memorization behaviors, we apply the complete two-stage pipeline to both code-specialized (StarCoder2-3B/7B) and general-purpose (Llama3-8B) models. As defined in our methodology, we measure successful extractions using Exact Match (EM) and BLEU-4 scores between the generated completions and the ground-truth suffixes. Table 1 (left columns) summarizes the overall extraction rates.

**Table 1: Overall and Categorical Extraction Rates Across LLMs**

| Model | Overall | | Categorical Extraction Rate (%) | | | | |
|---|---|---|---|---|---|---|---|
| | EM (%) | BLEU-4 | License | Docs | Dicts | Code | Testing |
| StarCoder2-3B | 64.16 | 0.848 | 72 | 64 | 62 | 39 | 22 |
| StarCoder2-7B | 62.00 | 0.836 | 70 | 63 | 63 | 39 | 21 |
| LLaMA 3-8B | 42.24 | 0.821 | 45 | 20 | 55 | 26 | 11 |

**Memorization persists across modern LLMs.** All three models exhibited substantial memorization, with EM rates ranging from 42.24% to 64.16%. This confirms that the pipeline designed for earlier models [7] remains effective for state-of-the-art LLMs, and that unintended memorization continues to be an intrinsic behavior of these systems.

**Code-specialized models memorize more training data.** The StarCoder2 variants showed significantly higher extraction rates (62-64%) compared to Llama3-8B (42%). This difference likely stems from their training objectives: StarCoder2 models are trained exclusively on curated code corpora (The Stack v2), whereas Llama3 is trained on diverse data including natural language. This specialization appears to amplify memorization of domain-specific content, as the models encounter repeated code patterns more frequently during training.

**Parameter size has limited impact at this scale.** Interestingly, increasing parameters from 3B to 7B in StarCoder2 did not correspond to higher memorization rates—both models demonstrated comparable extraction rates. This suggests that at this scale, architectural choices and training data characteristics may dominate over parameter count in determining memorization behavior. This finding contrasts with earlier work showing monotonic increases in memorization with model size [3], suggesting that the relationship between scale and memorization may be more nuanced in modern architectures.

> **RQ1 Findings:** Memorization persists in state-of-the-art code LLMs, with rates varying more by training domain than by parameter count. The existing pipeline remains effective for revealing memorization in recent models.

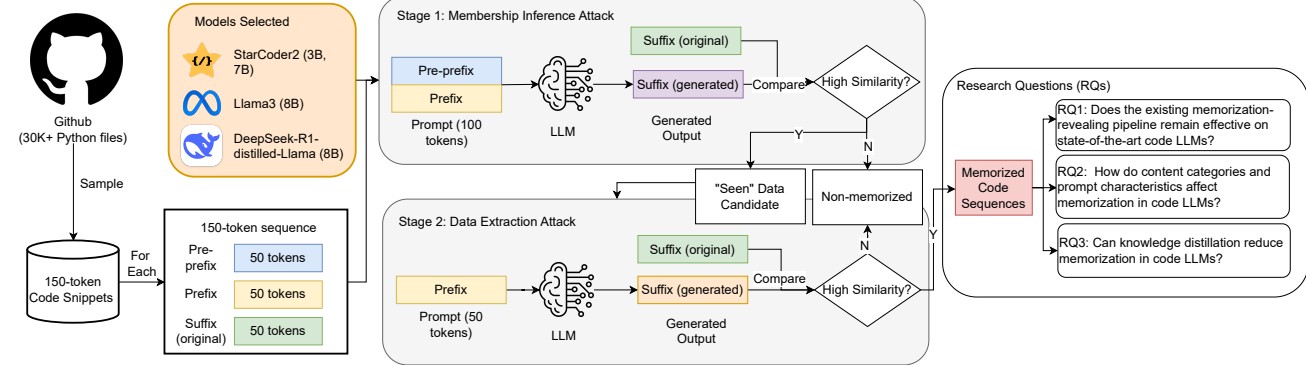

Figure 1: Overview of this research.

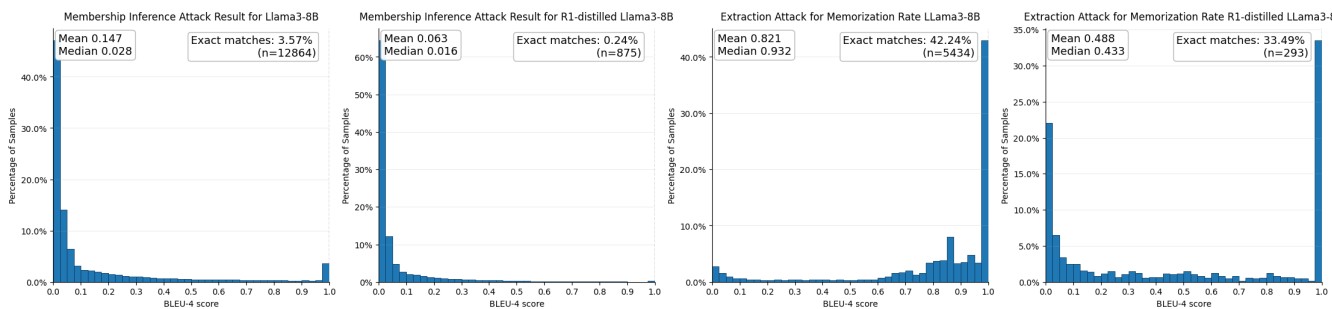

Figure 2: Extraction Attack for Memorization Rate R1-distilled LLama3-8B

## RQ2: How do content categories and prompt characteristics affect memorization in code LLMs?

We investigate how memorization varies across different types of code content and whether realistic developer workflows can trigger unintentional data leakage.

**1) Categorical Memorization.** Following prior work [7], we categorized memorized sequences into five functional types: license headers, documentation, data structures (dicts), code logic, and testing code. Table 1 details these categorical extraction rates.

**Repetitive boilerplate content exhibits the highest memorization rates.** Across both StarCoder2 models, license headers exhibited the highest memorization rates (~70%), closely followed by documentation. This aligns with the mechanics of LLM training: legal boilerplates and standardized docstrings act as highly predictable anchors across thousands of GitHub repositories, causing models to heavily overfit these sequences.

**Data structures pose a severe privacy threat due to high extraction rates.** An alarming finding is the high extraction rate of data structures (Dicts), particularly in Llama3-8B (55%). Since dictionaries frequently store configuration mappings, hardcoded credentials, or lookup tables, this high memorization rate exposes a severe attack vector for extracting sensitive project-specific data or Personally Identifiable Information (PII).

**Complex testing code resists memorization despite its repetitive framework syntax.** Interestingly, testing code exhibited the absolute lowest memorization rates (11-22%). While framework syntax (e.g., `pytest`, `JUnit`) is repetitive, actual assertions, mock data, and unit logic are hyper-specific to individual repositories. This high structural variability prevents models from effectively memorizing exact test implementations.

**2) Unintentional Memorization in Realistic Scenarios.** Standard data extraction attacks rely on arbitrary fixed-length prefixes, which do not reflect real-world usage. To assess the true threat to developers, we investigated whether benign, realistic prompts could trigger the same memorization. We simulated standard code completion tasks by replacing the artificial 50-token prefix with natural prompts, such as a function signature followed by a docstring.

**Realistic prompts trigger substantial unintentional memorization.** Without any adversarial intent or malicious prompting, StarCoder2-7B unintentionally reproduced exact training data (BLEU > 0.85) in 13.8% of its completions. Llama3-8B exhibited similar behavior, leaking exact data in 12.7% of cases.

**AI coding assistants inadvertently turn developers into accidental plagiarists.** These rates are highly concerning because they represent an invisible threat in modern software engineering workflows. The prevalence of unintentional memorization suggests that developers using AI coding assistants (e.g., GitHub Copilot) could inadvertently insert licensed, copyrighted, or proprietary code snippets into their own codebases simply by writing standard function descriptions. This exposes both individual developers and organizations to significant intellectual property violations and licensing compliance risks without their knowledge.

> **RQ2 Findings:** Memorization is strongly dictated by content structure: boilerplates (licenses) and data structures (dicts) are highly vulnerable, whereas highly coupled logic (tests) resists memorization. Alarmingly, normal developer workflows trigger unintentional verbatim leakage in 13-14% of cases, posing a severe compliance trap.

## RQ3: Can knowledge distillation reduce memorization in code LLMs?

To evaluate distillation as a mitigation strategy, we subjected Llama3-8B and its distilled variant, DeepSeek-R1-distilled-Llama-8B, to identical two-stage extraction pipelines. Since both share the same base parameters and training dataset, distillation is isolated as the primary variable.

**Distillation significantly reduces extraction rates.** The distilled model exhibited a notably lower extraction rate (34.17%) compared to the original Llama3-8B (42.24%), representing an approximate 19% reduction in memorized content. Notably, both models were trained on identical datasets, isolating distillation as the primary variable. This suggests that the distillation process itself can mitigate verbatim recall.

**Distillation affects membership inference detectability.** As we observe in Figure 2, the distilled model generated far fewer positive matches in membership inference attacks. This indicates that sequences present in the training set are less likely to be identified through standard MIA techniques after distillation. This finding suggests that distillation may force the model to "unlearn" specific training instances in favor of generalized patterns.

**Distillation is cost-effective but incomplete.** Unlike resource-intensive defenses such as differential privacy or dataset deduplication, knowledge distillation is straightforward to implement and often yields faster, more efficient models. However, the distilled model still exhibited a 34% extraction rate, indicating that distillation does not eliminate memorization entirely. This suggests that distillation should be considered as one component in a multi-layered mitigation strategy rather than a standalone solution.

> **RQ3 Findings:** Knowledge distillation reduces memorization by approximately 19% in Llama3-8B, offering a cost-effective mitigation approach. However, distillation alone does not eliminate memorization and should be combined with other defenses for comprehensive protection.

## Implications

Our findings carry important implications for both researchers and practitioners working with code LLMs.

**For researchers:** Our results confirm that memorization remains a persistent challenge in modern LLMs and is influenced by multiple factors beyond parameter count, including training domain and data characteristics. Future work should explore categorical memorization across diverse datasets and develop quantitative metrics for assessing memorization risk at different granularities. Additionally, investigating the mechanisms by which distillation reduces memorization could inform the design of more effective mitigation strategies.

**For practitioners:** The prevalence of unintentional memorization poses practical risks for AI-assisted coding tools. Developers and tool providers should implement stricter review processes for model outputs, particularly when generated code may contain licensed or proprietary content. Organizations deploying code LLMs should consider adopting the memorization-revealing pipeline as part of their validation and auditing workflows. Furthermore, incorporating knowledge distillation into model deployment pipelines could reduce memorization risks while maintaining performance.

**For policymakers:** The potential for LLMs to inadvertently leak licensed code or sensitive information raises legal and ethical concerns. Clear guidelines are needed regarding the use of memorized training data in AI-generated outputs, particularly in commercial settings. Providers should be transparent about memorization risks and implement technical safeguards to minimize unintended data exposure.

## 5 Threats to validity

**Internal validity.** While we compared models of different sizes (3B, 7B, 8B), these models also differ in architecture and training procedures. StarCoder2 models are trained exclusively on code, while Llama3-8B is trained on diverse data. We cannot fully disentangle the effects of parameter count, training data composition, and architectural design. Our knowledge distillation evaluation compared only one teacher-student pair (DeepSeek-R1 and Llama3-8B), and the observed reduction may be specific to this configuration.

**External validity.** We evaluated models up to 8 billion parameters, which is relatively small compared to industrial models with hundreds of billions of parameters (e.g., GPT-4, Claude 3). Our observations may not hold at larger scales. We focused exclusively on open-source models and Python code from public GitHub repositories. Memorization patterns may differ for proprietary LLMs, other programming languages, or codebases with different structural characteristics.

**Construct validity.** There is no universally accepted definition of memorization in language models. We operationalize memorization through exact match and BLEU-4 scores, but this may not capture all forms such as semantic patterns or paraphrased content. The threshold for determining memorization is somewhat arbitrary and could be adjusted, potentially changing our conclusions. BLEU-4 measures n-gram overlap and may not fully capture subtle memorization forms like structurally similar code with variable renaming. Our categorization of code into five types relies on heuristic rules and may introduce classification errors.

## 6 Conclusion

This study investigates memorization behavior in state-of-the-art code LLMs through a systematic two-stage attack pipeline. We find that memorization persists across modern models, with extraction rates ranging from 42% to 64%, confirming that the existing pipeline remains effective for recent LLMs. Our categorical analysis reveals that repetitive content (e.g., license headers and documentation) exhibits the highest memorization rates, while complex code such as testing logic is less susceptible. Realistic code completion scenarios can trigger unintentional memorization at rates of 13-14%, raising practical concerns for AI-assisted development tools. Our investigation of knowledge distillation demonstrates promising results, with the distilled Llama3-8B showing an approximate 19% reduction in extraction rates. These findings establish that memorization is influenced more by training domain and content characteristics than by parameter count alone, and that distillation offers a cost-effective mitigation approach, though incomplete as a standalone solution.

**Data Availability** The experimental data and scripts are available at https://figshare.com/s/e04567825b93bf02f62c.

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
