# OpenReview forum: "When AI Coding Assistants Leak Training Data: Study Memorization in Code LLMs"
_ACM.org/AIWare/2026/Conference — AIware 2026_

### Official Review · Reviewer_9xs2 · 2026-03-11

**Rating:** 4
**Confidence:** 3

**Review:**

### Pros:
- Practically relevant topic (although a number of studies already exist)
- Well-structured RQ progression, i.e., from pipeline validation to categorical analysis to mitigation
- The investigation of unintentional memorization under realistic code completion scenarios is a nice
- Replication package is available
---
### Cons (kindly see my detailed comments below):
- The knowledge distillation experiment may be biased (?), as DeepSeek-R1-distilled-Llama-8B differs from Llama3-8B in more ways than just distillation
- The evaluation scale is limited (models up to 8B and Python only); the study could benefit from including more LLMs.
- Some key experimental details need clarification, such as membership inference ground truth validation, threshold justification, and statistical testing (like p-values)
---
### Detailed Comments
Overall, this paper addresses an important problem, although some prior work exists and the paper reads somewhat as confirmation of known findings. That said, the analysis still provides useful insights, which is good for a short paper. The following may strengthen the work:

My primary concern is the experimental design of RQ3. The paper claims that distillation is isolated as the primary variable because Llama3-8B and DeepSeek-R1-distilled-Llama-8B "share the same base parameters and training dataset." However, the DeepSeek-R1 distillation process involves reinforcement learning from reasoning traces and a reasoning-oriented training objective, introducing some addtions well beyond standard distillation. Thus the observed reduction in extraction rate may not be attributed to distillation alone. Maybe the authors can either (a) perform distillation themselves under controlled conditions (e.g., distill StarCoder2-7B into a 3B student on identical data), or (b) explicitly enumerate other factors and mitigate the claims.

Also, some technical details could be made clearer. First, the membership inference stage lacks ground truth validation. The authors sample from BigQuery dataset and assume overlap with the model training data -- The Stack v2 for StarCoder2, which is indeed known, but, an undisclosed mix for Llama3 -- yet never quantify this overlap. So there is no confirmation that the test sequences are actually in the training set, the reported memorization rates may not be true. At least, the authors should verify a subset of flagged sequences against the Stack-v2 data. Also, the choice of 50-token granularity not well justified. The authors can discuss how sensitive the reported rates are to this threshold.

In addition, one further point -- the paper evaluates models up to 8B parameters and python only. A brief discussion of whether memorization trends observed at this scale are expected to hold at larger models would strengthen the paper.

**Summary:**

This paper investigates memorization behavior in code LLMs. The authors present a two-stage attack pipeline that combines membership inference with data extraction, and use it to evaluate 4 LLMs (StarCoder2-3B, StarCoder2-7B, Llama3-8B, and DeepSeek-R1-distilled-Llama-8B) on a dataset of 30,000+ Python files. The main findings include that memorization rates range from 42–64%, with code-specialized models memorizing more than general-purpose ones, and that repetitive content like licenses and documentation is most susceptible, while test code is least susceptible, at rates up to 70%. Also, realistic code completion prompts trigger unintentional memorization in 13–14% of cases, posing practical risks for AI coding assistants, and knowledge distillation reduces extraction rates by approximately 19%. The paper confirms that memorization is driven more by training domain and content characteristics than by parameter count, and positions distillation as a cost-effective but incomplete mitigation strategy.

---

> ### Author Response · Authors · 2026-03-19
> **Response to Reviewer 9xs2**
>
> Dear Reviewer 9xs2,
>
> We sincerely thank you for feedback and for recognizing the practical relevance of our realistic scenario experiments. Your constructive feedback regarding our experimental design and evaluation scope is highly appreciated.
>
> Below is our point-by-point response to your specific concerns and suggestions:
>
> 1. **Potential Bias in the Knowledge Distillation Experiment (Cons #1)**
>
>     **Response:** We completely agree. Claiming to have "isolated distillation as the primary variable" was an overstatement that ignored the profound impact of the reasoning-oriented teacher distribution and the reinforcement learning objective.
>
>     In the revised manuscript, rather than claiming standard distillation universally mitigates memorization, we will narrow our claims to state that *this specific reasoning-oriented distilled model* exhibits a lower extraction rate under our evaluation pipeline.
>
> 2. **Evaluation Scale and Scope (Cons #2)**
>
>     **Response:** Thank you for the insights. We acknowledge that our focus on open-weight models under 8B parameters and a Python-only dataset limits the external validity of our findings regarding scale. We will expand the "Threats to Validity" section to explicitly discuss expected scaling behaviours based on recent literature. Additionally, we will frame our current setup as a scoping constraint and note that empirically testing larger models (e.g., CodeLlama-13B) is part of a future extension work.
>
> 3. **MIA Ground Truth Validation and Thresholds (Cons #3)**
>
>     **Response:** We completely agree that these details are crucial for the reproducibility and rigor of our findings. In the revised manuscript, we will address all three areas:
>
>     - *Ground truth*: We will reframe Stage 1 as “Similarity-based Extractability Filter”. We will also add a paragraph to the "Threats to Validity" section discussing the lack of cryptographic ground-truth verification and the resulting risks of false positives/negatives.
>     - *Threshold justification:* We will justify the 50-token threshold (aligning with prior benchmarking standards to provide sufficient syntactic context while remaining short enough to test exact recall) and discuss how sensitive the reported rates might be to this parameter.
>     - Statistical testing: We apologize for this omission. We will perform and report appropriate statistical significance tests (e.g., Chi-square tests for comparing extraction rate proportions between models/categories, and p-values for the reduction achieved by distillation) to rigorously validate that our observed differences are statistically significant.
>
> Thank you once again for your constructive review and the support of our research.
>
> Sincerely,
>
> Authors of Submission #38

---

> > ### Comment · Reviewer_9xs2 · 2026-03-20
> >
> > Thank you for the detailed response and I very appreciate the willingness to address the concerns like the revised framing of the distillation claims and the planned addition of statistical tests. As my original rating was already `Accept', I will maintain my score. Hopefully the paper can be accepted, and also looking forward to seeing the improvements in the revised manuscript.

---

### Official Review · Reviewer_Yr6n · 2026-03-11

**Rating:** 4
**Confidence:** 4

**Review:**

**Pros**
- The paper addresses a relevant and timely topic.
- The manuscript is generally well written and well organized.
- The evaluation includes models from multiple LLM families.

**Cons**
- The novelty of the work is not sufficiently articulated.
- The investigation of knowledge distillation does not align well with established unlearning practice.
- The limitations of the dataset construction process are not properly acknowledged.

**Evaluation**


1. In my view, the main limitation of this paper is its novelty. The authors argue that prior work has mostly focused on older-generation language models, whereas this paper studies newer models such as StarCoder and Llama. However, recent work by Salerno et al. (2025) has already evaluated similar models for this task. Another aspect presented as novel is the analysis of the impact of knowledge distillation. However, I do not think this aspect is investigated convincingly enough (see next comment). I encourage the authors to articulate more clearly what is genuinely new in their contribution.
    - F. Salerno, A. Al-Kaswan and M. Izadi, "How Much Do Code Language Models Remember? An Investigation on Data Extraction Attacks Before and After Fine-tuning," in 2025 IEEE/ACM 22nd International Conference on Mining Software Repositories (MSR)
2. The authors state that knowledge distillation has been employed to reduce memorization while preserving model capabilities, citing Dong et al. [10]. However, Dong et al. study self-distillation, where the same model acts as both teacher and student through logit adjustment. In contrast, the authors consider DeepSeek-R1-Distilled-Llama-8B, which corresponds to a substantially different setup. Specifically, this model is distilled from DeepSeek-R1, a reasoning model, into a student model based on Llama-8B. As a result, comparing Llama-8B with DeepSeek-R1-Distilled-Llama-8B may not isolate the effect of distillation itself, since lower memorization could instead stem from properties inherited from the teacher model. I encourage the authors to clarify this point.
3. To construct the dataset (stage 1), the authors follow a procedure similar to that of Al-Kaswan et al. [7]. This procedure relies on heuristics to determine which data points were presumably included in the training data. By definition, such a heuristic process is subject to limitations and potential sources of error. I therefore encourage the authors to discuss these limitations explicitly in the manuscript.
4. In the title, abstract, and introduction, the paper places strong emphasis on code LLMs. However, the empirical study seems to target both code-specialized and general-purpose LLMs. I would therefore encourage the authors to soften this emphasis and align the framing more closely with the actual scope of the study.
5. The authors often refer to the analyzed models as state-of-the-art. By current standards, I would not consider these models state-of-the-art. I therefore recommend reconsidering this characterization.
6. The authors may want to consider the following relevant related works:
    - F. Salerno, A. Al-Kaswan and M. Izadi, "How Much Do Code Language Models Remember? An Investigation on Data Extraction Attacks Before and After Fine-tuning," in 2025 IEEE/ACM 22nd International Conference on Mining Software Repositories (MSR)
    - Jiang, Y., Li, Z., Huang, S., Treude, C., Su, X., & Wang, T. (2025). Effective code membership inference for code completion models via adversarial prompts. arXiv preprint arXiv:2511.15107
    - Yuqing Nie, Chong Wang, Kailong Wang, Guoai Xu, Guosheng Xu, and Haoyu Wang. 2025. Decoding Secret Memorization in Code LLMs Through Token-Level Characterization. In Proceedings of the IEEE/ACM 47th International Conference on Software Engineering (ICSE '25).

**Summary:**

The paper investigates memorization in LLMs using the data extraction attack introduced by Al-Kaswan et al. [7]. The evaluation covers four models (StarCoder2-3B, StarCoder2-7B, Llama-3-8B, and DeepSeek-R1-Distilled-Llama-8B) on a custom dataset of more than 30,000 Python files.

---

> ### Author Response · Authors · 2026-03-19
> **Response to Reviewer Yr6n**
>
> Dear Reviewer Yr6n,
>
> We sincerely thank you for your constructive review and for suggesting highly relevant, recently published literature. Your feedback is instrumental to help us properly scope our claims and accurately position our work in the landscape of LLM memorization research.
>
> Below is a detailed point-by-point response to your concerns and suggestions:
>
> 1. **Articulating Novelty and Integrating Recent Works (Cons #1 & Evaluation #1, #5, #6)**
>
>     **Response:** We greatly appreciate you pointing out these excellent recent papers. We will cite and discuss them (Salerno et al., Jiang et al., Nie et al.) in our new revision. Furthermore, we will revise our introduction to precisely articulate our distinct and complementary novelty, as outlined below:
>
>     1. While Jiang et al. focus on targeted adversarial prompting, we highlight the severe frequency of unintentional data leakage triggered by non-adversarial, realistic developer workflows.
>     2. Nie et al. investigate token-level secret memorization, our study provides a functional categorical analysis that demonstrates how structural characteristics (e.g., dictionaries vs. complex testing logic) inherently dictate memorization vulnerability.
>     3. While Salerno et al. investigate fine-tuning dynamics, our study distinctly evaluates knowledge distillation as a potential mitigation strategy and explicitly compares the memorization profiles of general-purpose versus code-specialized architectures in coding tasks.
> 2. **The Knowledge Distillation Setup is Confounded (Cons #2 & Evaluation #2)**
>
>     **Response:** We completely agree that comparing Llama3-8B with DeepSeek-R1-distilled-Llama-8B introduces confounding variables (namely, the reasoning-oriented teacher distribution and the RL training objective). Our claim that we "isolated distillation as the primary variable" was inaccurate.
>
>     In the revision, we will clarify that we are observing a reduction in extraction rates for this specific reasoning-distilled model, rather than proving that standard knowledge distillation universally acts as an unlearning mechanism. We will also carefully distinguish our setup from the self-distillation approach used by Dong et al.
>
> 3. **Limitations of the Dataset Construction Process (Cons #3 & Evaluation #3)**
>
>     **Response:** Thanks for your insights and we completely agree that the dataset construction process has limitations.
>
>     In the revised manuscript, we will expand the "Threats to Validity" section to explicitly detail these limitations. Specifically, we will note the absence of cryptographic ground-truth verification (such as querying Data Portraits) and discuss the specific risks of **false positives** (e.g., flagging highly predictable boilerplate as memorized) and **false negatives** (e.g., failing to trigger memorized snippets due to suboptimal prompting).
>
> 4. **Emphasis on "Code LLMs" and Reconsidering "State-of-the-art" (Evaluation #4)**
>
>     **Response:** We appreciate your detailed observation. Indeed, Llama-3 was included to serve as a general-purpose baseline against the code-specialized models (e.g., StarCoder2). We will soften this emphasis throughout the abstract and introduction, clarifying that our study provides a comparative analysis between code-specialized models and general-purpose models applied to coding tasks.
>
>     We also agree that “state-of-the-art” becomes an overstatement given the current frontier models. We will conduct a global pass to replace the term with more accurate descriptors such as "recent open-weight models" or "widely adopted accessible models”.
>
>
> Thank you again for your time, expertise, and the highly relevant literature recommendations. We are confident that these revisions will make the paper significantly more rigorous and properly contextualized.
>
> Sincerely,
>
> Authors of Submission #38

---

> > ### Comment · Reviewer_Yr6n · 2026-03-20
> >
> > I thank the authors for their willingness to articulate the novelty of their work more clearly, clarify their knowledge distillation setup, and acknowledge the limitations of their dataset construction process. I will update my score to "accept".

---

### Official Review · Reviewer_cTdk · 2026-03-12

**Rating:** 3
**Confidence:** 4

**Review:**

Strength:
The paper tackles an important and timely problem. Memorization and training data leakage remain active concerns for LLM safety and legal compliance, particularly for code models used in developer tools.

The empirical study across several modern open-source models is useful, especially the comparison between code-specialized models and general LLMs, as well as the distilled vs. non-distilled setting.

I also found the experiments using realistic code completion prompts to be a practical addition. Showing that leakage can occur even without adversarial prompting makes the risk more relevant to real developer workflows.

Weaknesses:
The main limitation is novelty. The methodology largely follows existing memorization evaluation pipelines (membership inference + extraction). The contribution is mainly an empirical application of these techniques to relatively recent code models.

Some of the causal interpretations are also stronger than what the experiments support. For example, attributing higher memorization to the “training domain” or suggesting that distillation forces models to generalize is plausible but not directly validated. Differences between models could also arise from architecture, dataset composition, or training procedures.

Finally, the description of the dataset is somewhat brief. More details about repository selection, deduplication, and category labeling would improve clarity and reproducibility.


Suggestions

1.Including at least one larger code model (e.g., CodeLlama or DeepSeek-Coder) would help assess whether the findings hold beyond ≤8B models.

2. Provide more details on dataset construction and deduplication.

3. Tone down causal claims or support them with controlled experiments.

**Summary:**

This paper studies memorization and potential training data leakage in code LLMs. The authors evaluate several open models (StarCoder2-3B/7B, Llama3-8B, and DeepSeek-R1-distilled-Llama-8B) using a two-stage pipeline combining membership inference and data extraction attacks. The evaluation uses a dataset of ~30K Python files from GitHub, where each file is split into pre-prefix, prefix, and suffix segments to test whether models can reconstruct the suffix from partial context.

The main findings are: (1) extraction rates between 42–64% exact match, (2) higher memorization in code-specialized models compared to general LLMs, (3) stronger leakage for boilerplate content such as licenses and documentation, (4) non-adversarial prompts can reproduce training data in ~13–14% of cases, and (5) distillation reduces extraction rates by about 19%, though it does not eliminate memorization.

---

> ### Author Response · Authors · 2026-03-19
> **Response to Reviewer cTdk**
>
> Dear Reviewer cTdk,
>
> We sincerely thank you for recognizing the practical relevance of our study. Your suggestions for improving the clarity and rigor of our paper are highly appreciated.
>
> Below is a detailed point-by-point response to your concerns and suggestions:
>
> 1. **Toning Down Causal Interpretations (Weakness 2 & Suggestion 3)**
>
>     **Response:** We fully agree with your critique and appreciate the suggestions. Our current phrasing implies a strong causal link where our experiments primarily establish correlation.
>
>     In the revised manuscript, we will carefully tone down these causal claims. When discussing the differences between StarCoder2 and Llama3, we will explicitly acknowledge that varying architectures, dataset compositions, and distinct training procedures all contribute to the observed differences, rather than attributing it exclusively to the "training domain." Similarly, regarding knowledge distillation, we will rephrase our conclusions to present the reduced extraction rate as an empirical observation of this specific distilled model under our pipeline, rather than a proven mechanism that "forces" generalization.
>
> 2. **Providing More Details on Dataset Construction (Weakness 3 & Suggestion 2)**
>
>     **Response:** You are absolutely correct. We had to compress the dataset description due to the page limitations, which inevitably harmed the clarity and reproducibility. In the revision, we will try to include a detailed breakdown of the dataset construction. And for details over repository selection, deduplication, and category labeling, we will include the details in our replication package repository to ensure sufficient clarity and reproducibility.
>
> 3. **Including a Larger Code Model (Suggestion 1)**
>
>     Response: We appreciate your constructive suggestions that addresses our primary threat to external validity. Given the short rebuttal window, we plan to run our evaluation pipeline on larger open-weight models (e.g., CodeLlama-13B) as part of a future extension work. In the revised manuscript, we will prominently expand our "Threats to Validity" section to discuss the expected scaling behaviours based on recent literature, explicitly framing the ≤8B limit as a scoping constraint.
>
>
> Thank you once again for your constructive feedback. We believe these revisions directly address your concerns and significantly improve the paper's scientific rigor and reproducibility.
>
> Sincerely,
>
> Authors of Submission #38

---

### Official Review · Reviewer_QwDu · 2026-03-13

**Rating:** 3
**Confidence:** 4

**Review:**

The problem is important, and parts of the setup are useful, in particular, the category analysis and the attempt to test more realistic prompting scenarios could have practical value. But at it's current state I don’t think the experiments support the paper’s framing or its main conclusions.

My main concern is construct validity.

### Major Issues

**1. Stage 1 is not actually evaluated as a membership inference attack.**
The paper calls the first stage a "Membership Inference Attack" but it does not function as one. A membership inference attack is a binary classifier over training-set membership and must be evaluated with ground-truth labels to establish TPR and FPR [^1].

Here, Stage 1 prompts the model with 100 tokens and measures similarity between the generated continuation and the held-out suffix. Stage 2 then repeats the same basic procedure with a 50-token prompt. In other words, the two stages are not independent attacks on different properties; Stage 1 is a screening heuristic for extractability.

This matters because StarCoder2 is trained on The Stack v2, which is not just publicly available but also is queryable [^2] though [Data Portraits](https://stack-v2.dataportraits.org). I.e. there is a feasible way to compute actual membership labels and report real MIA performance metrics. Without verified membership or a proper negative control, the reported 42-64% figures are not well supported as memorization rates. At most, they are similarity-based extraction rates under this pipeline.

The authors cite Al-Kaswan et al. (2024) as the source of this pipeline, but in that work the 100-token prompting step is used as a filtering procedure to build an extractability benchmark, not as a validated membership inference attack. They explicitly acknowledge that they cannot confirm training-set membership for code models they are using (CodeGen training data provenance is not clear) and flag this as a limitation. They also include implicit negative controls (GPT-2, trained on WebText rather than the Pile, that shows near-zero extraction on Pile-derived samples).

This paper removes those caveats, relabels the filtering step as an "MIA" and then draws stronger conclusions than the setup supports.

**2. The distillation claim is confounded**
The paper compares Llama3-8B with DeepSeek-R1-distilled-Llama-8B and claim to "isolate distillation as the primary variable." I don’t think that follows. The distilled model differs not just by distillation as an abstract technique, but also by teacher distribution, training objective, and likely post-training behavior. A lower extraction rate in this comparison does not identify distillation itself as the cause.
This matters because the paper goes beyond describing a model-to-model difference and claims that distillation is a cost-effective mitigation that preserves performance. The submission does not provide a controlled teacher-student experiment, and it does not report utility, latency, or cost measurements. The current evidence supports a narrower claim: this particular distilled model shows lower extraction under this evaluation pipeline.

#### Minor Issues
The "realistic scenario" experiment is one of the paper’s most practically relevant results, but it receives only one paragraph. The paper gives very little detail about how these prompts were constructed, which files they came from, or how many examples were used. That makes the 13-14% figure hard to interpret or reproduce. There also seems to be inconsistency about the "realistic scenario" leakage rate: the introduction reports 26-43%, while the abstract and later results section (RQ2 finding) reports 13-14%.

Some of the discussion language is stronger than the evidence seem to warrant. For example, statements about developers becoming “accidental plagiarists” or dictionaries posing a “severe privacy threat” are better framed as possible implications, not as direct empirical conclusions from the presented experiments.

More generally, the paper repeatedly uses language that is stronger than its evidence. For example, it treats BLEU > 0.85 as "exact data" in the realistic-scenario experiment, and it repeatedly refers to the reported figures as memorization rates even though the membership assumption is not validated.

Overall, I would reject the paper in its current form. I would be open to a revision, but only if the authors either (1) validate Stage 1 as an actual membership inference attack with verified membership labels and standard MIA metrics, or (2) reframe the method as a similarity-based extractability benchmark and narrow the downstream claims accordingly. The distillation section likewise needs either a properly controlled comparison or a much more cautious interpretation.



[^1]: [Membership Inference Attacks From First Principles](https://arxiv.org/abs/2112.03570)
[^2]: [Data Portraits: Recording Foundation Model Training Data](https://arxiv.org/abs/2303.03919)

**Summary:**

This paper studies memorization in 4 code LLMs using a two-stage attack pipeline (similarity-based “membership inference” stage followed by a suffix extraction stage) and reports substantial memorization rates, analyzes memorization across five code categories, and evaluates knowledge distillation as a mitigation all using 30k random 150 token sequences from BigQuery dataset.

---

> ### Author Response · Authors · 2026-03-19
> **Response to Reviewer QwDu**
>
> Dear Reviewer QwDu,
>
> We sincerely thank you for your thorough and constructive review. We agree with your assessment that our terminology and claims were stronger than the experimental setup support.
>
> Below is a detailed point-by-point response to your concerns and suggestions:
>
> 1. **Stage 1 is not actually evaluated as a membership inference attack (Major Issue #1)**
>
>     **Response:** We completely agree with your critique. Without querying Data Portraits to establish a rigorous ground-truth membership (and thereby calculating true TPR/FPR0, labeling Stage 1 as membership inference attack is conceptually inaccurate.
>
>     In the revised manuscript, we will abandon the “MIA” terminology. We will reframe stage 1 as a “Similarity-based Extractability Filter”, which shall accurately reflect its role as a screening heuristic. Consequently, we will replace all mentions of "memorization rates" with the more precise **"similarity-based extraction rates."** We will also reinstate the caveats explicitly acknowledged by Al-Kaswan et al. (2024) to ensure proper contextualization of this pipeline's limitations.
>
> 2. **The distillation claim is confounded (Major Issue #2)**
>
>     **Response:** You are absolute correct. Our original claim of “isolated distillation as the primary variable", without isolating other impact factors, was an overstatement.
>
>     We will narrow our claims regarding distillation in the revision. We will revise the text to state specifically that *this particular reasoning-oriented distilled model* exhibits lower extraction rates under our evaluation pipeline. We will remove the generalized claim that standard knowledge distillation is a proven, standalone mitigation strategy, and instead present this finding as an observation of this specific model family's behavior.
>
> 3. **Inconsistency and Lack of Detail in "Realistic Scenarios" (Minor Issue 1)**
>
>     **Response**: We sincerely apologize for the confusion and the typo in the Introduction. The correct number is indeed 13%-14%, as presented in Abstract and Result sections. We will correct the typo in the Introduction.
>
>     Additionally, we will expand the methodology section to explicitly detail the the prompt construction for realistic scenarios.
>
> 4. **Overly Strong Language (Minor Issue 2)**
>
>     **Response:** We agree that our phrasing is over strong over the evidence warrants. We will conduct a global pass to tone down the language. Specifically,
>
>     - "Accidental plagiarists" will be changed to "potential exposure to licensing compliance risks."
>     - "Severe privacy threat" will be softened to "potential vector for sensitive data exposure."
>     - We will stop referring to BLEU > 0.85 as "exact data" and instead use "highly similar output."
>
> Thank you again for the feedback, which has undeniably made this paper more precise and reliable. We hope these explicit commitments to reframe the paper address your primary concerns.
>
> Sincerely,
>
> Authors of Submission #38

---

> > ### Comment · Reviewer_QwDu · 2026-03-20
> >
> > thank you for acknowledging the issues and the clarification on how they are going to be resolved in camera-ready version.
> >
> > Taking into consideration responsiveness and the proposed changes I’ll update the review rating to “weak accept”